# Integrated approach of brachial-ankle pulse wave velocity and cardiovascular risk scores for predicting the risk of cardiovascular events

**Hyue Mee Kim**[1], **Tae-Min Rhee**[2], **Hack-Lyoung Kim**[2]*

1 Division of Cardiology, Department of Internal Medicine, Chung-Ang University Hospital, Chung-Ang University College of Medicine, Seoul, South Korea, 2 Division of Cardiology, Department of Internal Medicine, Boramae Medical Center, Seoul National University College of Medicine, Seoul, South Korea

* khl2876@gmail.com

## Abstract

**Data Availability Statement:** All relevant data are within the paper.

**Funding:** The authors received no specific funding for this work.

### Background

The 2013 American College of Cardiology (ACC)/American Heart Association (AHA) atherosclerotic cardiovascular disease (ASCVD) risk score may be insufficient for accurate prediction of major adverse cardiac events (MACE) in Asians. This study was performed to investigate whether brachial-ankle pulse wave velocity (baPWV) has additional prognostic value to the risk score estimated by the ACC/AHA pooled cohort equations (PCEs).

### Methods

A total of 6,359 patients (3,534 men and 2,825 women) aged 40–79 years without documented cardiovascular disease who underwent baPWV measurement were retrospectively analyzed. Cardiovascular risk scores were calculated using the 2013 ACC/AHA PCEs. Cardiovascular events, including cardiac death, non-fatal myocardial infarction, coronary revascularization and ischemic stroke, were assessed.

### Results

During a median follow-up period of 4.0 years (interquartile range 1.7–6.1 years), cardiovascular events occurred in 129 patients (2.0%). The receiver operating characteristic curve analysis showed that baPWV was stronger in the detection of cardiovascular events than the 2013 ACC/AHA risk score (area under the curve: 0.70 versus 0.62, $p < 0.001$). In the multivariable Cox regression analysis, both baPWV and 2013 ACC/AHA risk score were independently associated with the occurrence of clinical events ($p < 0.001$ for each). The baPWV had incremental prognostic value to the 2013 ACC/AHA risk score in predicting clinical events (global chi-square from 21.23 to 49.51, $p < 0.001$).

**Competing interests:** The authors have declared that no competing interests exist.

## Conclusion

The baPWV appears to be a strong predictor of the risk of cardiovascular events in Koreans. Measuring baPWV in addition to the 2013 ACC/AHA risk score helps identify individuals at risk for MACE aged 40–79 years without previous cardiovascular diseases.

## Introduction

Atherosclerotic cardiovascular disease (ASCVD), including coronary heart disease and ischemic stroke, constitutes a major public health problem, showing high prevalence and mortality worldwide [1]. Controlling modifiable risk factors, such as hypertension, diabetes mellitus, dyslipidemia, smoking, physical inactivity and obesity, can effectively prevent the development of ASCVD [2, 3]. Additionally, it has been suggested that the benefits of risk management are great in individuals at high risk [4]. Therefore, early screening of high-risk patients and customized management are advantageous in terms of improving prognosis.

To predict ASCVD risk, many investigators have developed several risk-predicting algorithms [5–7]. In 2013, the American Heart Association (AHA) and the American College of Cardiology (ACC) announced a new ASCVD risk score calculated from age, sex, race, cholesterol levels, blood pressure, diabetes, smoking status and medication, to guide ASCVD risk-reducing treatments [4]. However, the applicability of this risk score to other races has been questioned, because it has been found to overestimate or underestimate the risk in several populations including East Asians [8–11].

Arterial stiffness, another important risk-predicting marker that reflects structural and functional changes in the arterial wall, is associated with the development of cardiovascular events independent of traditional risk factors [12–15]. Among various measures of arterial stiffness, brachial-ankle pulse wave velocity (baPWV) has recently been widely used due to its non-invasiveness and simplicity [16]. Moreover, clinical usefulness of baPWV has been validated in many studies [17–21].

Previously, our study group reported significant correlations between baPWV and several cardiovascular risk scores including 2013 ACC/AHA ASCVD risk score [22]. The aim of this study is to compare prognostic value between baPWV and 2013 ACC/AHA ASCVD risk score, and to determine whether baPWV has incremental value to 2013 ACC/AHA ASCVD risk score in predicting future cardiovascular events.

## Methods

### Study population

We retrospectively identified consecutive individuals who visited cardiovascular center of a general hospital in a big city (Seoul, South Korea) and underwent baPWV measurement as part of the cardiovascular examination between January 2010 and June 2018. A total of 11,767 individuals who were screened for the study. Among the individuals, we excluded the following patients: (1) those with a history of previous ASCVD, (2) those aged <40 years or >79 years, (3) those with missing information for calculating cardiovascular risk, and (4) those with a low ankle-brachial index (ABI <0.9) score. A total of 6,359 individuals were included in the final analysis. The study was carried out according to the principles of the Declaration of Helsinki and approved by the Clinical Research Institute (IRB) of Seoul National University

Boramae Medical Center (Seoul, South Korea). Informed consent was waived by the IRB due to its retrospective study design and routine nature of information collected.

## Clinical data collection

Body mass index (BMI) was calculated as weight (kg) divided by the square of the height (m$^2$). BMI $\geq$25 kg/m$^2$ was defined as obesity [23]. Systolic and diastolic blood pressures (SBP and DBP) were measured using an oscillometric device by a trained nurse. Hypertension was defined based on previous diagnosis, current use of anti-hypertensive medications or SBP/ DBP $\geq$ 140/90 mmHg. Diabetes mellitus was defined based on previous diagnosis, current use of anti-diabetic medications or fasting glucose $\geq$ 126 mg/dL. Dyslipidemia was defined based on previous diagnosis or low-density lipoprotein (LDL) cholesterol level $\geq$ 160 mg/dL [24]. Individuals were classified as smokers if they had smoked regularly during the previous 12 months. After overnight fasting, venous blood sample was obtained and blood levels of the following parameters were analyzed: total cholesterol, LDL cholesterol, high-density lipoprotein (HDL) cholesterol, triglyceride, C-reactive protein, hemoglobin, glycated hemoglobin and creatinine. Estimated glomerular filtration rate (GFR) was obtained by the Modification of Diet in Renal Disease (MDRD) equation. Concomitant cardiovascular medications were also assessed, which included antiplatelets, calcium channel blockers, renin-angiotensin system blockers, beta-blockers, and statin.

## baPWV measurement

We measured baPWV noninvasively using validated protocols that have previously been published [18, 20]. Briefly, caffeine ingestion and cigarette smoking were not allowed on the day of the measurement; however, regular medication was continued. After the patient was placed in the supine position for 5 or more minutes, blood pressure and baPWV were automatically generated using a noninvasive automated waveform analyzer (VP-1000; Colin Co., Ltd., Komaki, Japan). Then, baPWV was measured in upper and lower extremities with a plethysmographic sensor, which simultaneously recorded blood pressure, electrocardiogram, and heart sounds. The baPWV was calculated by dividing the distance by the transit time, and the distance between measurement points was estimated by individual height. The transit time was calculated from the starting point of the brachial pulse wave to the start of the ankle pulse wave. The mean values of right-sided and left-sided baPWV measurements were used for the analysis. All measurements were taken by an experienced operator, and the intra-observer coefficient of variation was approximately 5.1% in our laboratory [20].

## Calculation of the ACC/AHA risk scores

We calculated the 10-year ASCVD risk using the 2013 ACC/AHA pooled cohort equations (PCEs) [4]. We used sex-specific equations for non-Hispanic whites because there was no specific equation for Koreans. Risk scores were calculated using the following parameters: age, sex, hypertension, diabetes mellitus, smoking, total and HDL cholesterol and blood pressure. Cardiovascular risks of study subjects were classified according to the ACC/AHA risk scores: low risk, <5%; borderline risk, $\geq$5%—<7.5%; intermediate risk, $\geq$7.5%—<20%; or high risk, $\geq$20% [25].

## Assessment of cardiovascular events

Data on clinical events were obtained from hospital records described by physicians, telephone contracts or national death data. The primary endpoint of this study was a composite of the

occurrence of cardiac death, non-fatal myocardial infarction, coronary revascularization and ischemic stroke. Cardiac death was defined as one after acute coronary syndrome, pump failure with heart failure with reduced ejection fraction, stroke or sudden cardiac death. Non-fatal myocardial infarction was defined based on electrocardiographic findings, elevated cardiac enzyme levels and coronary angiography results. Coronary revascularization included percutaneous coronary intervention and/or coronary bypass surgery. Ischemic stroke was diagnosed by typical neurological signs and symptoms that were assessed by neurologists and noninvasive brain imaging findings, such as brain computed tomography and/or magnetic resonance imaging.

## Statistical analysis

Continuous variables are expressed as means ± standard deviations, and categorical variables are expressed as numbers and percentages. Differences between continuous variables were compared using Student's *t* test for independent variables, and differences between categorical variables were compared using Pearson's chi-square test. The cutoff value of baPWV to predict cardiovascular events was set according to the receiver operating characteristic (ROC) curve analysis with the highest sum of sensitivity and specificity. ROC curves were generated to compare the predictive value of baPWV and the ACC/AHA risk score. Comparisons of the areas under the curve (AUCs) were conducted using Delong's method [26]. Event rates were estimated using event counts and exposure over time. Cox proportional hazard regression analyses were performed to evaluate the predictive values of baPWV and ACC/AHA risk score. The following clinical covariates including age, sex, BMI, heart rate, diabetes, hypertension, dyslipidemia, smoking, beta blockers, RAS blockers, and statin were considered potential confounders and included as confounding variables in the multivariable analysis. The hazard ratios (HRs) and 95% confidence intervals (CIs) were calculated. Event-free survival analyses were conducted using the Kaplan–Meier method with the log-rank test and Cox proportional hazard model. All statistical analyses were performed using SPSS (version 22.0; SPSS Inc., Chicago, IL, USA), and statistical significance was set at $p < 0.05$.

## Results

### Baseline characteristics

The baseline characteristics of the study population are summarized in Table 1. The mean age of the study population was 59.9±8.8 years, and 55.6% of the patients were male. The prevalence rates of hypertension, diabetes mellitus, dyslipidemia and current smoking among the patients were 43.4%, 22.7%, 48.6%, and 13.8%, respectively. Approximately half of the patients (47.6%) had obesity. Major laboratory examination findings, including cholesterol, hemoglobin, and C-reactive protein levels and the GFR, were within the normal range. A total of 28.4% of the individuals took a renin-angiotensin system blocker, and 18.6% were prescribed beta-blockers. The proportion of patients taking antiplatelet drugs, calcium channel blockers and statins were 21.9%, 22.3% and 44.5%, respectively.

### Comparison between patients with and without clinical events

During a median follow-up period of 4.0 years (interquartile range 1.7–6.1 years), 129 cardiovascular events, including cardiovascular death (n = 10), non-fatal myocardial infarction (n = 2), coronary revascularization (n = 69) and ischemic stroke (n = 48), occurred. Comparisons of baseline characteristics between individuals with and without events are shown in Table 1. Patients with cardiovascular events were older and had more cardiovascular risk

**Table 1. Baseline characteristics of study population.**

| Characteristic | Total (n = 6359) | Subjects with events (n = 129) | Subjects without events (n = 6230) | *p* value |
|---|---|---|---|---|
| Age, years | 59.9±8.8 | 62.2±7.9 | 59.9±8.8 | 0.003 |
| Male, sex | 3534 (55.6) | 78 (60.5) | 3456 (55.5) | 0.259 |
| Body mass index, kg/m$^2$ | 25.0±3.3 | 25.0±3.0 | 25.0±3.3 | 0.815 |
| Systolic blood pressure, mmHg | 129.9±17.7 | 130.9±19.6 | 130.0±17.7 | 0.534 |
| Diastolic blood pressure, mmHg | 78.3±11.0 | 76.8±10.8 | 78.4±11.1 | 0.124 |
| Heart rate, per min | 69.5±11.8 | 68.5±11.7 | 69.5±11.8 | 0.400 |
| Risk factors | | | | |
| Hypertension | 2758 (43.4) | 73 (56.6) | 2685 (43.1) | 0.002 |
| Diabetes mellitus | 1444 (22.7) | 49 (38.0) | 1395 (22.4) | <0.001 |
| Dyslipidemia | 3088 (48.6) | 90 (69.8) | 2998 (48.1) | <0.001 |
| Smoking | 878 (13.8) | 34 (26.4) | 844 (13.5) | <0.001 |
| Obesity (body mass index ≥25kg/m$^2$) | 3025 (47.6) | 58 (45.0) | 2967 (47.6) | 0.548 |
| Laboratory findings | | | | |
| Total cholesterol, mg/dL | 164.2±37.5 | 152.5±38.6 | 164.4±37.4 | <0.001 |
| Low-density lipoprotein cholesterol, mg/dL | 94.1±33.5 | 83.8±36.0 | 94.3±33.4 | <0.001 |
| High-density lipoprotein cholesterol, mg/dL | 49.9±12.8 | 47.5±14.6 | 50.0±12.8 | 0.032 |
| Triglyceride, mg/dL | 129.2±79.5 | 126.2±73.7 | 129.2±79.7 | 0.657 |
| C-reactive protein, mg/dL | 0.7±2.9 | 1.8±5.0 | 0.7±2.8 | <0.001 |
| Hemoglobin, g/dL | 13.8±1.7 | 13.3±1.8 | 13.8±1.7 | 0.005 |
| Glycated hemoglobin, % | 6.4±1.1 | 6.6±1.2 | 6.4±1.1 | 0.053 |
| Glomerular filtration rate, mL/min/1.73m$^2$ | 87.8±23.7 | 84.6±27.8 | 87.9±23.6 | 0.127 |
| Concomitant medications | | | | |
| Antiplatelet | 1394 (21.9) | 48 (37.2) | 1346 (21.6) | <0.001 |
| Calcium channel blocker | 1417 (22.3) | 31 (24.0) | 1386 (22.2) | 0.630 |
| RAS blocker | 1804 (28.4) | 46 (35.7) | 1758 (28.2) | 0.064 |
| Beta-blocker | 1184 (18.6) | 43 (33.3) | 1141 (17.9) | <0.001 |
| Statin | 2831 (44.5) | 78 (60.5) | 2753 (44.2) | <0.001 |

Numbers are expressed as mean±SD or n (%). RAS: renin-angiotensin system.

factors, including hypertension, diabetes mellitus, dyslipidemia and current smoking, compared to those without. Total cholesterol and LDL cholesterol levels were lower in patients with events than those without, and the other laboratory findings, including HDL cholesterol, triglyceride, and HbA1c levels and the GFR, were not significantly different between the 2 groups. The proportion of patients taking antiplatelet agents, beta-blockers and statins were higher in patients with events than in those without.

## Comparison of the predictive value between the ACC/AHA risk score and baPWV

The predictive value for cardiovascular events was compared between ACC/AHA risk score and baPWV using ROC curve analyses (Fig 1). The AUC of the ACC/AHA risk score and baPWV for predicting cardiovascular events were 0.62 (95% CI, 0.61–0.64, *p* <0.001) and 0.70 (95%, CI 0.69–0.71, *p* <0.001), respectively. Pairwise comparisons of the ROC curve showed that baPWV was more powerful in predicting cardiovascular events than the ACC/AHA risk score (*p* <0.001) (Table 2). In ROC curve analysis, the cutoff value of baPWV (= 1556 cm/s) for the best prediction of cardiovascular events was obtained. In Kaplan-Meier survival

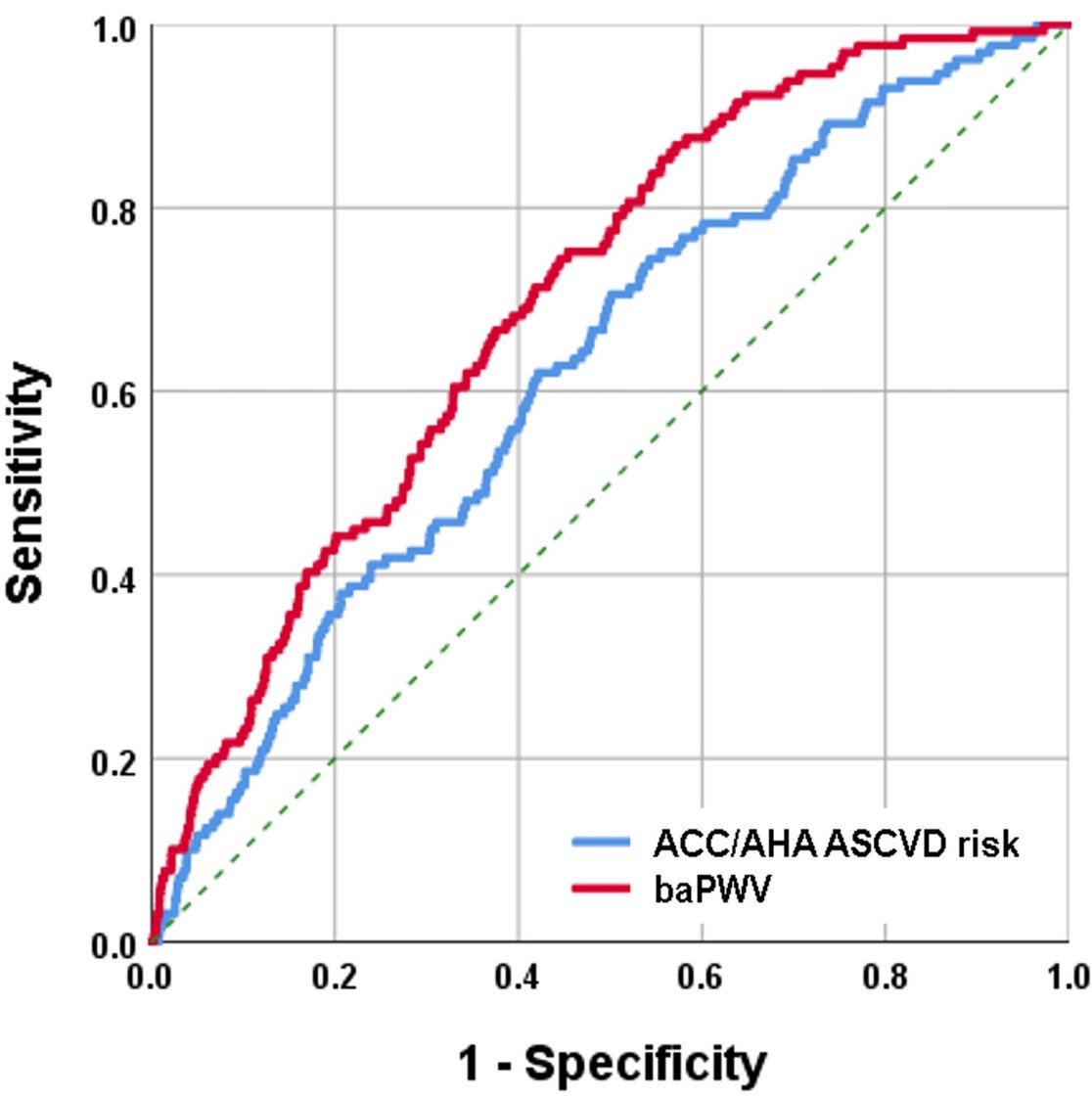

**Fig 1. The receiver operating characteristic analysis showing the prognostic value of ACC/AHA ASCVD risk score and baPWV.** ACC/AHA ASCVD, American College of Cardiology/American Heart Association atherosclerotic cardiovascular disease; baPWV, brachial-ankle pulse wave velocity.

**Table 2. Receiver operating characteristic curve analyses.**

| Parameter | AUC | 95% CI | $p$ value | Comparison with DeLong |
|---|---|---|---|---|
| 2013 ACC/AHA ASCVD risk score | 0.62 | 0.61–0.63 | <0.001 | Ref. |
| baPWV | 0.70 | 0.68–0.71 | <0.001 | 0.001 |

2013 ACC/AHA ASVD risk score: 2013 American College of Cardiology/American Heart Association atherosclerotic cardiovascular disease risk score, baPWV: brachial-ankle pulse wave velocity, AUC: area under the curve.

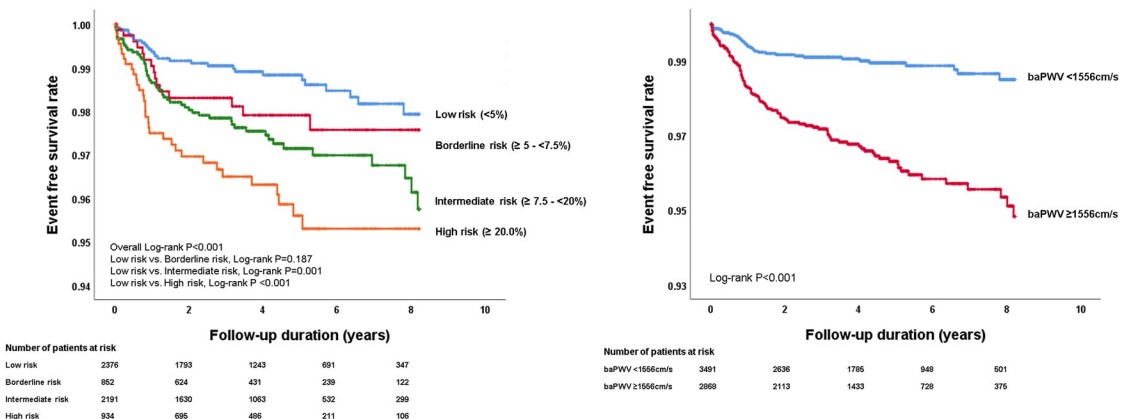

**Fig 2. Event-free survival rates according to ACC/AHA ASCVD risk score (A) and baPWV (B).** ACC/AHA ASCVD, American College of Cardiology/American Heart Association atherosclerotic cardiovascular disease; baPWV, brachial-ankle pulse wave velocity.

analysis, event-free survival rates were significantly different according to the cutoff values of baPWV and ACC/AHA risk scores (log-rank $p$ <0.001 for each) (Fig 2). In multivariable Cox regression analysis, patients with an intermediate ACC/AHA ASCVD risk (HR, 2.09; 95% CI, 1.32–3.30; $p$ = 0.002) and a high ACC/AHA ASCVD risk (HR, 3.02; 95% CI, 1.82–5.00; $p$ <0.001) had higher cardiovascular event rates compared to those with low ACC/AHA ASCVD risk. A higher baPWV (≥1,556 cm/s) was also significantly associated with an increased risk of cardiovascular events in the same multivariable analysis (HR, 3.75; 95% CI, 2.34–6.00; $p$ <0.001) (Table 3).

## Additional prognostic value of baPWV to the ACC/AHA risk score

A higher baPWV (≥1,556 cm/s) provided incremental prognostic value to the ACC/AHA risk score in the prediction of ASCVD (global chi-square score, from 21.23 to 49.51, $p$ <0.001) (Fig 3).

## Discussion

Our study demonstrated that both baPWV and the 2013 ACC/AHA risk score were independently associated with the development of future cardiovascular events in Korean subjects

**Table 3. Prognostic values of 2013ACC/AHA risk score and baPWV in predicting cardiovascular events.**

| Risk | HR | 95% CI | $p$ value |
|---|---|---|---|
| 2013 ACC/AHA ASCVD low risk (<5%) | 1 | | |
| 2013 ACC/AHA ASCVD Borderline risk (≥5–<7.5%) | 1.52 | 0.81–2.84 | 0.190 |
| 2013 ACC/AHA ASCVD Intermediate risk (≥7.5–<20%) | 2.09 | 1.32–3.30 | 0.002 |
| 2013 ACC/AHA ASCVD High risk (≥20%) | 3.02 | 1.82–5.00 | <0.001 |
| baPWV≥1,556 cm/s (unadjusted) | 3.58 | 2.41–5.31 | <0.001 |
| baPWV ≥1,556 cm/s (adjusted)* | 3.75 | 2.34–6.00 | <0.001 |

2013 ACC/AHA ASVD risk score: 2013 American College of Cardiology/American Heart Association atherosclerotic cardiovascular disease risk score, baPWV: brachial-ankle pulse wave velocity, HR: hazard ratio, CI: confidence interval.

*Covariates were controlled as potential confounders: Age, Sex, BMI, HR, DM, HTN, Dyslipidemia, Smoking, Beta blockers, RAS blockers, Statin.

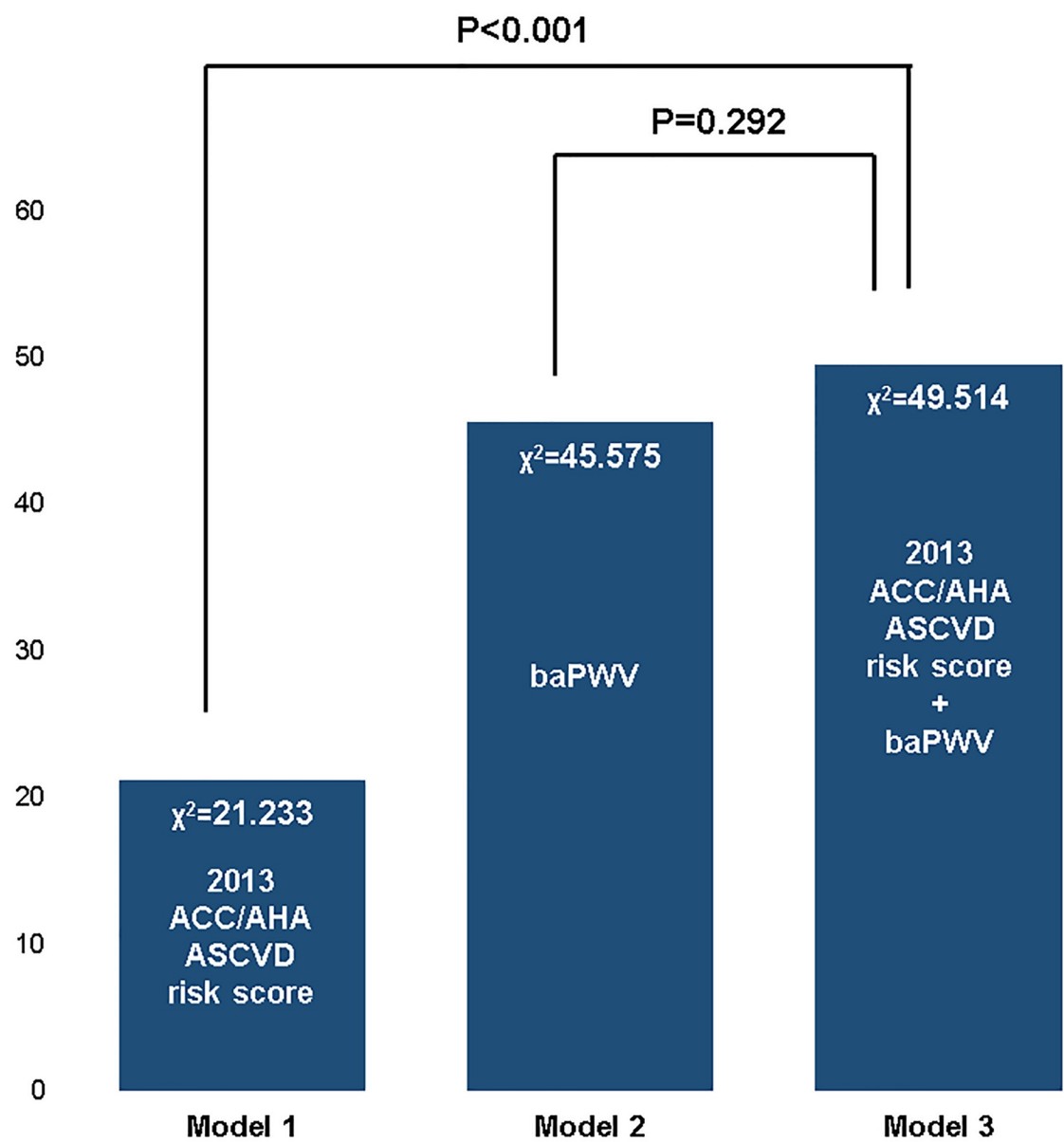

**Fig 3. The additive prognostic value of baPWV.** ACC/AHA, American College of Cardiology/American Heart Association; baPWV, brachial-ankle pulse wave velocity.

aged 40–79 years without prior history of documented cardiovascular disease. The baPWV showed a stronger association with the risk of cardiovascular events than the ACC/AHA risk score. In addition, high baPWV provided incremental prognostic value over the ACC/AHA risk score, regarding the prediction of cardiovascular events. To the best of our knowledge, this is the first study to show the prognostic value of baPWV compared to the ACC/AHA risk score. Furthermore, our findings suggest for the first time that an integrated approach of baPWV and ACC/AHA risk score could improve risk assessment.

The 2013 ACC/AHA PCEs were derived using data from 5 large epidemiological cohort studies conducted in the US (Atherosclerosis Risk in Communities, Cardiovascular Health Study, Coronary Artery Risk Development in Young Adults and the Framingham and

Framingham Offspring studies); however, the data for the follow-up of Hispanic and Asian-American cohorts were limited [4]. Many countries in Asia and Europe have performed external validation of the risk score system, and they found overestimated or underestimated cardiovascular risk by the PCEs in each country [8–11]. Although several countries have developed their own risk prediction model for ASCVD [27, 28], they are under-utilized, and the 2013 ACC/AHA PCEs is still widely used due to its accessibility. In this background, further testing to complement the ACC/AHA PCEs for appropriate risk prediction in these countries would be needed. Given our findings, we believe that non-invasive and easy to measure baPWV will be a suitable test for this purpose.

Arterial stiffness is the result of the degenerative process affected by aging and arteriosclerosis [29, 30]. Measuring arterial stiffness is clinically important because it provides prognostic implications for cardiovascular diseases [12, 31]. PWV is the most widely used measure of arterial stiffness [16, 32]. As PWV is a measure of time it takes the pulse to travel between the 2 arterial points, it is a direct measure of arterial stiffness. PWV is proportional to the speed of pulse wave through the arterial walls: when the blood vessels are young and elastic, the speed of the pulse wave traveling along the arterial wall is low; otherwise, in stiffened artery, the PWV is high [22]. Although carotid-femoral pulse wave velocity(cfPWV) is considered a gold standard non-invasive measure of arteria stiffness and has abundant clinical data [16, 32], it has several shortcomings such as technical difficulty and inconvenience to the patients during the exam [16]. Palpation of the carotid and femoral pulses in obese patients is difficult and makes patients uncomfortable. To overcome these limitations, baPWV has been introduced. baPWV is easier and simpler to measure than cfPWV, and is widely used in Asians [16]. Although there are several criticisms of baPWV, including its inclusion of large peripheral muscular arteries, the limitation on the distance predicted by height, and underestimation in patients with peripheral artery stenosis, aortic stenosis, or aneurysm [33, 34], the usefulness of baPWV has also been validated by many clinical studies [17–20] and meta-analyses [35, 36]. Measuring baPWV is recommended by the guidelines for the management of hypertension in Japan [37]. Considering its non-invasiveness and simplicity, baPWV may be a good risk stratification tool especially in mass screening.

Most newly eligible individuals who need risk stratification have no known cardiovascular diseases and are at low risk. Recently, achievement of various studies regarding ASCVD preventive effects of statins have broadened the spectrum of statins in primary prevention [25]. Therefore, it is clinically important to select individuals for whom the statins can be helpful by appropriating risk stratification considering the benefits and harm or side effects caused by long-term treatment of statins. In general, clinical risk assessment could improve when using 2 or more risk stratification tools rather than using 1 method. Previous studies showed that the addition of baPWV to clinical and laboratory findings significantly increased the predictive power of future cardiovascular events [13, 20, 38, 39]. Additionally, we confirmed the additional predictive value of baPWV for ASCVD when it added to the ACC/AHA PCEs in this study. In line with our findings, coronary calcium scoring (CACS) have recently been recommended for helping guide decisions about preventive intervention in select individuals [25]. Given that CACS can reclassify estimated risk and modify treatment plan when it is added to risk scores, our study suggests that baPWV could also be used as an additive tool to more meticulously stratify cardiovascular risk in primary prevention.

## Study limitations

This study has several limitations. First, this was a single-center retrospective study; thus, there might be a selection bias. Secondly, as our data were from Koreans aged 40–79 years,

generalizing the results in other races is challenging. Thirdly, we used the 2013 ACC/AHA PCEs for non-Hispanic whites; thus, there would be discrepancies in applying the algorithms to Koreans. Fourthly, baPWV measurement may not be available in all centers as a part of routine clinical practice because it needs a specific vender and well-trained operators. Finally, baPWV is known to be affected by other cardiovascular risk factors. Although we excluded patients with a low ABI score and made corrections using the multivariable analysis, we could not completely rule out possible confounding factors.

## Conclusions

The prognostic value of baPWV for future cardiovascular events in the Korean general population was greater than that of the 2013 ACC/AHA PCEs. Also, combined information on baPWV and the 2013 ACC/AHA PCEs showed additive predictive value. Our findings suggest that an integrated approach of baPWV and the ACC/AHA risk scores could improve risk assessment and management.

## Author Contributions

**Conceptualization:** Hyue Mee Kim, Hack-Lyoung Kim.

**Data curation:** Hyue Mee Kim, Tae-Min Rhee, Hack-Lyoung Kim.

**Formal analysis:** Hyue Mee Kim, Tae-Min Rhee, Hack-Lyoung Kim.

**Investigation:** Hyue Mee Kim, Tae-Min Rhee, Hack-Lyoung Kim.

**Methodology:** Hyue Mee Kim, Hack-Lyoung Kim.

**Project administration:** Hack-Lyoung Kim.

**Resources:** Hack-Lyoung Kim.

**Software:** Hyue Mee Kim, Hack-Lyoung Kim.

**Supervision:** Hack-Lyoung Kim.

**Validation:** Hyue Mee Kim, Tae-Min Rhee, Hack-Lyoung Kim.

**Visualization:** Hyue Mee Kim, Tae-Min Rhee, Hack-Lyoung Kim.

**Writing – original draft:** Hyue Mee Kim.

**Writing – review & editing:** Hyue Mee Kim, Tae-Min Rhee, Hack-Lyoung Kim.

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
