## [Decision Letter · Decision Letter 0]

24 Mar 2022

PONE-D-22-03869Integrated Approach of Brachial-Ankle Pulse Wave Velocity and Cardiovascular Risk Scores for Predicting the Risk of Cardiovascular EventsPLOS ONE

Dear Dr. KIm,

Thank you for submitting your manuscript to PLOS ONE. After careful consideration, we feel that it has merit but does not fully meet PLOS ONE’s publication criteria as it currently stands. Therefore, we invite you to submit a revised version of the manuscript that addresses the points raised during the review process.

Please submit your revised manuscript by May 16  If you will need more time than this to complete your revisions, please reply to this message or contact the journal office at plosone@plos.org. Please include the following items when submitting your revised manuscript:A rebuttal letter that responds to each point raised by the academic editor and reviewer(s). You should upload this letter as a separate file labeled 'Response to Reviewers'.A marked-up copy of your manuscript that highlights changes made to the original version. You should upload this as a separate file labeled 'Revised Manuscript with Track Changes'.An unmarked version of your revised paper without tracked changes. You should upload this as a separate file labeled 'Manuscript'.

We look forward to receiving your revised manuscript.

Kind regards,

Timir Paul

Academic Editor

PLOS ONE

**Please respond to the following Comments:**

**Editor Comments :**

Kim HM et al evaluated the Integrated Approach of Brachial-Ankle Pulse Wave Velocity and Cardiovascular Risk Scores for Predicting the Risk of Cardiovascular Events. It has shown that baPWV has further predictive value compared to the ACC/AHA risk score and have superior predictive ability when used in combination. Sample size is adequate.

I have following specific comments:

Measuring baPWV would be challenging and there is a higher chance of measurement error, need a specific vendor, well trained technician/sonographer. It is relatively costly and time consuming and may not be readily available in all centers. These limitations should be included in details in the limitation section.Need to provide more references on these bapvw, validity of aortic pwv, carotid femoral pwv and their limitations. Discussing why those are not favorite options what advantages and disadvantages.I would mention the current ESC and ACC/AHA risk prediction guideline and their recommendations on the measurement of the arterial stiffness.The ACC/AHA 2010 guidelines give CLASS III: NO BENEFIT recommendation for Specific Measures of Arterial Stiffness Measures of arterial stiffness outside of research settings are not recommended for cardiovascular risk assessment in asymptomatic adults. (Level of Evidence: C) However Measurement of Ankle-Brachial Index is CLASS IIa .  Measurement of ankle-brachial index is reasonable for cardiovascular risk assessment in asymptomatic adults at intermediate risk (Level of Evidence: B). Certainly the ABI is the easiest to calculate, need little time and without cost as just need to take ankle and brachial BP.  I would recommend to compare the ROC between ABI and baPwv and add separate analysis with ROC using ABI and PECs in the same population. That would be more valuable if possible. You have excluded ABI < 0.9. what about to include those and compare as suggested.

General comments:

Abstract Methods:  “A total of 6,359 patients (3,534 men and 2,825 women) aged 40-9 years without..” , it would be 79 instead of 9Definition of hypertension used is old “Hypertension was defined based on previous diagnosis, current use of anti-hypertensive medications or SBP/DBP ≥ 140/90 mmHg” It has changed many years back. The definition would be <130/80Obesity defined as (body mass index ≥25kg/m2 ), it would be overweight and obesity defined as BMI > 30“baPWV could also be used as an addictive tool to more meticulously stratify cardiovascular risk in primary prevention” Not addictive tool it would be additive tool.

Statistical analyses:

“The following clinical covariates were considered 137 potential confounders and included as confounding variables in the multivariable analysis”. But there are no variables mentioned

**Reviewers' comments:**

**Reviewer #1: **Brachial-Ankle Pulse Wave Velocity:" Is it a risk factor or modifier?"

It's an interesting thought provoking retrospective study. Dr.Kim's work laid a stepping stone to fill in the gap to estimate ASCVD risk in non-Caucasian population. We need more prospective studies like these to consider adding ba-PWV in ASCVD risk assessment tool.

Reviewer's Responses to Questions

**Comments to the Author**

1. Is the manuscript technically sound, and do the data support the conclusions?

Reviewer #1: Yes

2. Has the statistical analysis been performed appropriately and rigorously? 

Reviewer #1: Yes

3. Have the authors made all data underlying the findings in their manuscript fully available?

Reviewer #1: No

4. Is the manuscript presented in an intelligible fashion and written in standard English?

Reviewer #1: Yes

5. Review Comments to the Author

Reviewer #1: Brachial-Ankle Pulse Wave Velocity:" Is it a risk factor or modifier?"

It's an interesting thought provoking retrospective study. Dr.Kim's work laid a stepping stone to fill in the gap to estimate ASCVD risk in non-Caucasian population. We need more prospective studies like these to consider adding ba-PWV in ASCVD risk assessment tool.

6. PLOS authors have the option to publish the peer review history of their article (what does this mean?). If published, this will include your full peer review and any attached files.

Reviewer #1: **Yes: **Maheswara Satya Golla

---

## [Author Response · Author response to Decision Letter 0]

10 Apr 2022

Point-by-point Response

Responses to Editor Comments to the Author:

We sincerely thank Editor for his/her comments on our manuscript.

Comment #1

Measuring baPWV would be challenging and there is a higher chance of measurement error, need a specific vendor, well trained technician/sonographer. It is relatively costly and time consuming and may not be readily available in all centers. These limitations should be included in details in the limitation section.

Response to comment #1

We thank for this valuable comment. We agree with your comments that there are several limitations of PWV measurements such as requirement of specialized instrument and well-trained operators. We have added these limitations in the discussion section.

Changes for comment #1

We have revised the manuscript as below in Discussion section.

Discussion section: Fourthly, baPWV measurement may not be available in all centers as a part of routine clinical practice because it requires a specific vender and well-trained operators.

Comment #2

Need to provide more references on these bapvw, validity of aortic pwv, carotid femoral pwv and their limitations. Discussing why those are not favorite options what advantages and disadvantages.

Response to comment #2

We thank for this important comment. Carotid femoral PWV (cfPWV) and baPWV are the most widely used types of PWV measurement.1 cfPWV measurement were initiated earlier, thus it has a large amount of clinical data and is better validated. Additionally, cfPWV has been considered the gold standard measurement for large artery stiffness, because the aorta and its first branches are the most pathophysiologically important areas while considering arterial stiffness.2 However, measurement of cfPWV requires much skill and time for the placement of pressure-sensitive transducers on the carotid and femoral arteries, especially in obese subjects. Additionally, palpation of the femoral artery can be psychologically invasive to some subjects.1 To overcome these limitations, baPWV has been introduced. It has a procedural advantage of being simple to use, requiring only the wrapping of pressure cuffs. There are many clinical data showing baPWV can predictcardiovascular outcomes3-6. However, baPWV has been criticized for the following: including large peripheral muscular arteries, non validation of distance predicted by height, and underestimation in patients with peripheral artery stenosis, aortic aneurysm, and aortic stenosis.7 Therefore, specific exclusion criteria are needed in clinical research, hence we excluded those with ankle-brachial index <0.9, as generally suggested.8 

Changes for comment #2

We revised manuscript as below in Discussion section.

Discussion section: Although carotid-femoral pulse(cfPWV) wave velocity is considered a gold standard non-invasive measure of arteria stiffness and has abundant clinical data.[16, 32] it has several shortcomings such as technical difficulty and inconvenience to the patients during the exam.[16] Palpation of the carotid and femoral pulses in obese patients is difficult and makes patients uncomfortable. To overcome these limitations, baPWV has been introduced. baPWV is easier and simpler to measure than cfPWV, and is widely used in Asians.[16] Although there are several criticisms of baPWV, including its inclusion of large peripheral muscular arteries, the limitation on the distance predicted by height, and underestimation in patients with peripheral artery stenosis, aortic stenosis, or aneurysm[33, 34], the usefulness of baPWV has also been validated by many clinical studies[17-20] and meta-analyses.[35, 36]

Comment #3&#4

I would mention the current ESC and ACC/AHA risk prediction guideline and their recommendations on the measurement of the arterial stiffness.

The ACC/AHA 2010 guidelines give CLASS III: NO BENEFIT recommendation for Specific Measures of Arterial Stiffness Measures of arterial stiffness outside of research settings are not recommended for cardiovascular risk assessment in asymptomatic adults. (Level of Evidence: C) However Measurement of Ankle-Brachial Index is CLASS IIa. Measurement of ankle-brachial index is reasonable for cardiovascular risk assessment in asymptomatic adults at intermediate risk (Level of Evidence: B). Certainly the ABI is the easiest to calculate, need little time and without cost as just need to take ankle and brachial BP.

Response to comment #3 &#4

Thank you for your kind comments. Both the Ankle-brachial index (ABI) and the PWV are known to be associated with prevalent cardiovascular diseases, but they are different types of risk assessment tools. The ABI is an indicator of atherosclerosis in the legs.9 Atherosclerosis is the narrowing of the artery by the deposition of atheromatous plaque, typically occurring in the intima layer. The PWV is an indicator of arteriosclerosis and arterial stiffness, being one of the earliest changes in vascular structure. It is a result of degenerative processes in the vessel wall, mainly in the medial layer of large elastic arteries, which is distinct from atherosclerosis.10 

Generally, PWV measurement becomes unreliable if the circulation is disrupted on the way of pulse propagation. In this regard, bilateral measurements of the ABI are mandatory before measurement of baPWV to confirm the normal circulation of the lower limb arteries. When the ABI is lower than 0.9, peripheral artery disease (PAD) is suspected. These patients are considered to have high risk of cardiovascular disease and require closer investigation; but the value of baPWV is difficult to interpret in such situation. However, if the ABI is 0.9 or higher, PAD may be excluded.7 These patients have a higher risk of cardiovascular disease when the baPWV is elevated. In our study, subjects with ABI <0.9 were excluded, and baPWV ≥1,556 cm/s was an indicator of increased cardiovascular risk. In summary, PWV and ABI can be used complementarily, but it is difficult to compare the relative superiority of the two indicators.

Additionally, in the 2021 ESC guideline on cardiovascular prevention in clinical practice, which is the most recent guideline, both ABI and PWV were introduced as potential risk modifiers.11 

Comment #5

I would recommend to compare the ROC between ABI and baPwv and add separate analysis with ROC using ABI and PECs in the same population. That would be more valuable if possible. You have excluded ABI < 0.9. what about to include those and compare as suggested.

Response to comment #5

Thank you for your valuable comment. As you recommended, we analyzed with ROC using ABI in the same population. However, as I mentioned in the previous comment, patients with ABI <0.9 should be excluded in further baPWV study due to low reliability. Therefore, all the patients in our registry had ABI ≥0.9. 

The AUC of ABI (lower value between left and right) in our study was 0.55 (95% CI: 0.49-0.60, P=0.078). Pairwise comparisons of the ROC curve showed that baPWV had a more powerful predictive value in detecting ASCVD than the ABI (P <0.001 for comparison).

As we mentioned in comment 4, ABI is one of the good potential risk modifiers, but it is useful especially for patients with an abnormal ABI<0.9.12 Our study aimed to evaluate the prognostic value of PWV, thus new population are needed to evaluate the prognostic value of ABI. 

General comment #1

Abstract Methods: “A total of 6,359 patients (3,534 men and 2,825 women) aged 40-9 years without..” , it would be 79 instead of 9

Response to general comment #1

Thank you. We have amended the text from ‘9’ to ‘79’.

General comment #2

Definition of hypertension used is old “Hypertension was defined based on previous diagnosis, current use of anti-hypertensive medications or SBP/DBP ≥ 140/90 mmHg” It has changed many years back. The definition would be <130/80

Response to general comment #2

Thank you for your comments. As you mentioned, although the definition of hypertension in the ACC/AHA guidelines have been changed13, other guidelines, such as ESC, ISH, and Korean guidelines maintain the definition of hypertension as SBP/DBP≥140/90 mmHg14-16. Since most of the subjects in this study were Koreans, we think it is more appropriate to define hypertension as SBP/DBP ≥140/90 mmHg in this study.

General comment #3

Obesity defined as (body mass index ≥25kg/m2), it would be overweight and obesity defined as BMI > 30

Response to general comment #3

Thank you for your comment. The World Health Organization (WHO) defines obesity as BMI ≥30 kg/m2, but East Asians, including Koreans, have increased risk of cardiovascular disease even below a BMI of 25 kg/m2 and have higher body fat than Westerners at the same BMI.17 Therefore, in Asia pacific region, BMI of higher than 25 kg/m2 is considered as obesity.14 

General comment #4

“baPWV could also be used as an addictive tool to more meticulously stratify cardiovascular risk in primary prevention” Not addictive tool it would be additive tool.

Response to general comment #4

Thank you for your comments. We changed from addictive to additive.

General comment #5

“The following clinical covariates were considered 137 potential confounders and included as confounding variables in the multivariable analysis”. But there are no variables mentioned

Response to general comment #5

Previously, we mentioned confounding variables in Table 3. We have added the co variables in the Methods section.

Methods: The following clinical covariates, including age, sex, BMI, heart rate, diabetes, hypertension, dyslipidemia, smoking, beta blockers, RAS blockers, and statin were considered potential confounders and included as confounding variables in the multivariable analysis.  

Point-by-point Response

Responses Reviewer#1 Comments to the Author:

General comment

Brachial-Ankle Pulse Wave Velocity:" Is it a risk factor or modifier?"

It's an interesting thought provoking retrospective study. Dr.Kim's work laid a stepping stone to fill in the gap to estimate ASCVD risk in non-Caucasian population. We need more prospective studies like these to consider adding ba-PWV in ASCVD risk assessment tool.

Response to comment

Thank you for your kind comment.

---

## [Editor Report · Decision Letter 1]

12 Apr 2022

Integrated Approach of Brachial-Ankle Pulse Wave Velocity and Cardiovascular Risk Scores for Predicting the Risk of Cardiovascular Events

PONE-D-22-03869R1

Dear Dr. Kim,

We’re pleased to inform you that your manuscript has been judged scientifically suitable for publication and will be formally accepted for publication once it meets all outstanding technical requirements.

Kind regards,

Timir Paul

Academic Editor

PLOS ONE

Additional Editor Comments (optional):

All comments have been addressed.
---

## [Editor Report · Acceptance letter]

18 Apr 2022

PONE-D-22-03869R1 

Integrated Approach of Brachial-Ankle Pulse Wave Velocity and Cardiovascular Risk Scores for Predicting the Risk of Cardiovascular Events 

Dear Dr. Kim:

I'm pleased to inform you that your manuscript has been deemed suitable for publication in PLOS ONE. Congratulations! Your manuscript is now with our production department. 

Kind regards, 

on behalf of

Dr. Timir Paul 

Academic Editor

PLOS ONE